# Learning to Better Search with Language Models via Guided Reinforced Self-Training

**Seungyong Moon[1], Bumsoo Park[2], Hyun Oh Song[1]** *
[1]Seoul National University, [2]KRAFTON
{symoon11,hyunoh}@mllab.snu.ac.kr
bumsoo.park96@krafton.com

## Abstract

While language models have shown remarkable performance across diverse tasks, they still encounter challenges in complex reasoning scenarios. Recent research suggests that language models trained on linearized search traces toward solutions, rather than solely on the final solutions, exhibit improved generalization, despite the search traces being potentially noisy or suboptimal. However, relying on such imperfect traces can result in inefficient use of test-time compute. To address this, we propose *guided reinforced self-training* (Guided-ReST), a fine-tuning algorithm designed to improve the model's capability for effective search during inference. The key insight behind Guided-ReST is that optimal solutions can serve as valuable step-by-step landmarks to guide the model's search process. Based on this insight, we introduce a novel data generation method that seamlessly incorporates optimal solutions into the model's search procedure, enabling the generation of high-quality search traces. By fine-tuning the model on these search traces, we effectively distill improved search strategies into the model. Our method significantly enhances the search capabilities of language models on arithmetic reasoning and code self-repair tasks, including Countdown, CodeContests, and CodeForces. We release the source code at `https://github.com/snu-mllab/guided-rest`.

## 1   Introduction

Transformer-based language models have achieved remarkable success, demonstrating human-level performance across a wide range of natural language tasks, including conversation, code generation, and mathematical problem-solving [1, 31, 23, 12, 15, 26]. Their impressive performance is primarily attributed to auto-regressive training on massive, internet-sourced corpora. However, language models still encounter challenges in tasks that require complex planning and reasoning [18, 33]. To address these challenges, recent approaches have leveraged prompt-based strategies, enabling self-correction and planning via external symbolic search algorithms [34, 27, 38]. While these methods have shown success in specific contexts, their reliance on explicit prompting templates and manually engineered search structures restricts their flexibility and scalability.

More recent studies indicates that training language models to internalize the search process, rather than solely outputting the final solution, significantly improves their reasoning capabilities and allows performance to scale effectively with increased test-time compute [4, 9, 7]. An instructive example of test-time scaling is stream of search (SoS) [4], which trains a model to imitate search traces that encompass the complete decision-making process in finding optimal solutions through trial and error, including exploration and backtracking upon failure. This work demonstrates that models trained on search traces, despite being noisy and even suboptimal, exhibit superior generalization performance compared to those trained solely to imitate optimal solution via behavior cloning (BC) [22]. However,

---

* Corresponding author

39th Conference on Neural Information Processing Systems (NeurIPS 2025).

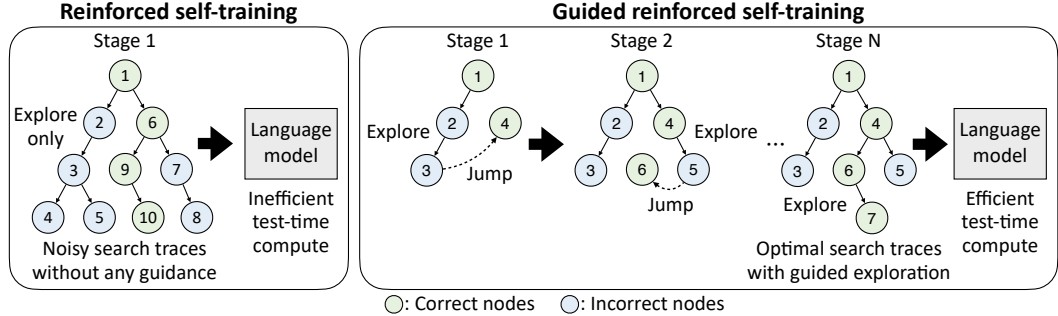

Figure 1: Overview of guided reinforced self-training, illustrating how search traces are generated through the progressive integration of optimal solutions during self-generation. The numbers indicate the order in which nodes are explored.

excessively noisy search traces can result in inefficient utilization of test-time compute, as the model may repeatedly explore irrelevant paths or perform redundant backtracking.

In this work, we examine how the sub-optimality of search traces affects test-time compute efficiency and propose a novel self-training algorithm, *guided reinforced self-training* (Guided-ReST), designed to improve search efficiency during inference. Our key insight is that while optimal solutions alone are insufficient for direct imitation, they can effectively serve as valuable landmarks to guide the search process. Inspired by jump-start reinforcement learning (JSRL) [32], we introduce a new data generation approach that progressively integrates optimal solutions into the model's search process, generating high-quality search traces, as illustrated in Figure 1. Subsequently, we fine-tune the model on these traces, thereby distilling more effective search strategies into the model. Finally, we further enhance performance by applying reinforcement learning (RL) fine-tuning.

We evaluate our method on the Countdown benchmark [4], a challenging arithmetic reasoning task that requires complex search. Our method achieves over a 10% accuracy improvement compared to baseline fine-tuning algorithms. Furthermore, it utilizes test-time compute more efficiently, reaching comparable performance while using less than half the number of tokens required by other baselines. Finally, we demonstrate the applicability of our method to a more realistic code self-repair task on the CodeContests and CodeForces benchmarks [14, 21].

## 2 Preliminaries

### 2.1 Stream of search

In this paper, we consider the problem of training a language model $\pi_\theta$ for tasks that require complex searching capabilities. Suppose that we have a training dataset $\mathcal{D}$ of question-solution pairs $(q, S)$, where each solution consists of step-by-step reasoning $S = (s_1, \ldots, s_T)$. The most straightforward approach to train the model on this dataset is behavior cloning (BC) [22], which optimizes the model to generate the optimal solution using supervised learning:

$$\max_\theta \mathbb{E}_{(q,S)\sim\mathcal{D}} \left[ \log \pi_\theta(S \mid q) \right].$$

However, prior studies have demonstrated that this approach struggles to generalize to unseen test examples [37, 11, 4, 39].

To address this, the steam of search (SoS) method introduces a novel training approach that emphasizes the search process itself rather than just the final solution [4]. Specifically, SoS formulates the problem as a tree search, exploring potential solutions through trial and error with operations until a successful solution is discovered. Each node in the tree represents a partial solution, and each edge represents a single reasoning step. The method represents primitive tree-search operations in language, including node generation, exploration, verification, and backtracking. For each question, it employs symbolic search algorithms such as depth-first search (DFS) and breadth-first search (BFS) to generate a search trace $Z = (z_1, \ldots, z_{T'})$, where each $z_t$ denotes an individual search operation expressed in language.

The model is then trained to predict this search trace via supervised learning:

$$\max_\theta \mathbb{E}_{(q,Z)\sim\mathcal{D}} \left[ \log \pi_\theta(Z \mid q) \right].$$

Note that in this framework, the optimal solution can also be viewed as a search trace, namely a path in the search tree. Additional details are provided in Appendix A.

## 2.2 Fine-tuning with self-generated data

While SoS demonstrates improved generalization by training a language model on search traces rather than optimal solutions, these traces are often noisy and suboptimal, resulting in imperfect alignment with the target task. To address this, SoS further fine-tunes the model using reinforcement learning techniques. One approach is reinforced self-training (ReST) [42, 6, 28], which fine-tunes the model on successful examples drawn from its self-generated responses via supervised learning:

$$\max_\theta \mathbb{E}_{q\sim\mathcal{D}, Z\sim\pi_\theta(\cdot|q)} \left[ \mathbb{1}_{R(Z|q)>\tau} \cdot \log \pi_\theta(Z \mid q) \right],$$

where $R$ denotes the terminal reward function that evaluates the quality of the responses and $\tau$ denotes the threshold parameter.

Another approach is reinforcement learning (RL) fine-tuning [30, 17], which fine-tunes the model to directly maximize the reward while constraining the KL divergence from the reference model $\pi_{\text{ref}}$ initialized from the pre-trained model:

$$\max_\theta \mathbb{E}_{q\sim\mathcal{D}, Z\sim\pi_\theta(\cdot|q)} \left[ R(Z \mid q) - \beta \cdot D_{\text{KL}}(\pi_\theta(\cdot \mid q) \parallel \pi_{\text{ref}}(\cdot \mid q)) \right],$$

where $\beta > 0$ denotes the coefficient controlling the strength of the KL penalty term. This objective is typically optimized with proximal policy optimization (PPO) or group-relative policy optimization (GRPO) [25, 26].

## 2.3 Countdown benchmark

We use Countdown as the primary benchmark, following previous studies on searching with language models [38, 4]. Each problem consists of input numbers and a target number, and the objective is to combine the inputs using the four basic arithmetic operations to reach the target. For example, given a target number 26 and input numbers $[84, 2, 14, 15]$, a valid solution is $26 = 84 - 2 \times (14 + 15)$. Each arithmetic operation in this solution can be considered a single reasoning step. This task involves a high branching factor up to $O(k^2)$ at each node of the search tree, where $k$ is the number of remaining inputs. To ensure a tractable difficulty level, we set the number of initial inputs to 4, consistent with the setup in Gandhi et al. [4]. Additional details are provided in Appendix B.

## 3 Methods

### 3.1 Motivation

SoS demonstrates that a language model trained on noisy, suboptimal search traces generalizes better than one trained only on clean, optimal solutions [4]. However, optimal solutions still offer valuable guidance during search. By providing the model with partial optimal solutions as hints, we can steer the model toward a reduced and more manageable search space, enabling more efficient exploration and significantly increasing the likelihood of success under a given token budget. Although optimal solutions are available only during training, the resulting trajectories provide high-quality supervision for fine-tuning.

This approach is closely aligned with the principle of jump-start reinforcement learning (JSRL) [32], which introduces a two-policy framework comprising a guide policy and an exploration policy. The guide policy, informed by prior knowledge or demonstrations, initiates the search process by steering the model toward promising regions of the solution space. The exploration policy then takes over and continues the search to reach the optimal solution, which significantly reduces the cost of exploration.

To investigate whether SoS can also benefit from this approach, we conduct a preliminary experiment. Given the optimal solution $S = (s_1, \ldots, s_T)$, we define a partial solution $(s_1, \ldots, s_t)$ consisting of

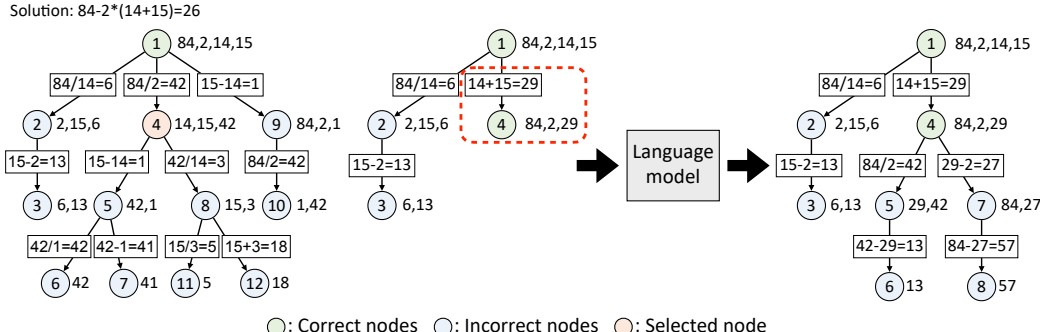

**Step 1**. Select a random child node from the last subgoal node.

**Step 2**. Replace the selected node to the next subgoal node and truncate the search trace up to it.

**Step 3**. Continue the search from the modified node.

◯ : Correct nodes   ◯ : Incorrect nodes   ◯ : Selected node

Figure 2: Overview of the subgoal augmentation algorithm. The numbers indicate the order in which nodes are explored. The red box highlights the modifications through node replacement.

Table 1: Accuracy and cross-entropy loss of search traces from partial solutions of different lengths.

| Length | 0 | 1 | 2 | 3 |
|---|---|---|---|---|
| Accuracy (%) ($\uparrow$) | 55.96 | 85.28 | 95.27 | 100.00 |
| Cross-entropy loss ($\downarrow$) | 0.0481 | 0.1245 | 0.2412 | 0.2539 |

the first $t$ reasoning steps. We utilize this partial solution as an initial prompt to condition the model in generating a response. Table 1 shows the accuracy of search traces initialized from partial solutions of different lengths, evaluated over 10K training examples. The model successfully continues the search and discovers solutions when starting from partial solutions, even though it has not encountered these specific partial solutions during training. Moreover, increasing the amount of guidance by providing longer partial solutions greatly improves the accuracy of the resulting traces. This motivates us to leverage such high-quality, self-generated traces for fine-tuning, enabling the model to effectively internalize the successful search strategies.

However, naively distilling partial solutions as hints might negatively impact the model's performance. Table 1 shows the cross-entropy loss of successful search traces initialized from partial solutions of different lengths, demonstrating that traces from longer partial solutions results in higher loss values. This happens because longer partial solutions follow a distribution that differs significantly from the model's search behavior. Consequently, fine-tuning directly on these traces might cause substantial parameter updates, potentially degrading the model's search capabilities. Therefore, it is important to explore methods for effectively integrating optimal solutions into self-generated traces, ensuring that the resulting traces maintain both high likelihood and solution quality.

### 3.2 Guided reinforced self-training

In this subsection, we introduce a novel fine-tuning algorithm, called *guided reinforced self-training* (Guided-ReST), which seamlessly incorporates optimal solutions into the model's search process for data generation and effectively distills improved search behaviors into the model. The main idea is to leverage unsuccessful search attempts as context for each intermediate reasoning step within the optimal solution, mimicking the model's inherent search process and resulting in trajectories with high likelihood under the model. Before delving into our approach, we first introduce some notations. We define a subgoal node as any node along the optimal solution in the search tree. Thus, the optimal solution consisting of $T$ reasoning steps contains $T$ subgoal nodes.

Now, consider an unsuccessful search trace that has reached the $(t-1)$-th subgoal node but fails to navigate its subtree, and thus does not reach the $t$-th subgoal node. To transform it into a successful search trace, our method first randomly selects an unsuccessfully explored child node of the $(t-1)$-th subgoal node. Next, it replaces this child node with the correct $k$-th subgoal node and truncate the

---

**Algorithm 1** AUGMENTSUBGOAL: Generating search traces via subgoal augmentation

---

**Require:** model $\pi_\theta$, question $q$, search trace $Z$, optimal solution $S$, subgoal index $t$
  1: **if** $Z$ has already explored the $t$-th subgoal node of $S$ **then**
  2:     **return** $Z$
  3: **end if**
  4: Randomly select an explored child node from the $(t-1)$-th subgoal node of $S$
  5: Replace the selected child node with the $t$-th subgoal of $S$
  6: Truncate the search trace up to the modified node
  7: Continue the search: $Z \leftarrow \pi_\theta(\cdot \mid q, Z)$
  8: **return** $Z$

---

---

**Algorithm 2** Guided reinforced self-training

---

**Require:** model $\pi_{\text{ref}}$, question $q$, optimal solution $S$
  1: Initialize model parameters: $\pi_\theta \leftarrow \pi_{\text{ref}}$
  2: **for** $i = 1, 2, \ldots, \text{MAXITER}$ **do**
  3:     Generate a search trace: $Z \leftarrow \pi_\theta(\cdot \mid q)$
  4:     **for** $t = 1, 2, \ldots, T$ **do**
  5:         **if** $Z$ is successful **then**
  6:             **break**
  7:         **end if**
  8:         Add the $t$-th subgoal to the search trace: $Z \leftarrow$ AUGMENTSUBGOAL$(\pi_\theta, q, Z, S, t)$
  9:     **end for**
 10:     Fine-tune the model via supervised learning: $\pi_\theta \leftarrow \text{TRAIN}(\pi_{\text{ref}}, q, Z)$
 11: **end for**

---

search trace up to the modified node, as the subsequent search history becomes inconsistent with the updated subgoal. After the modification, the model continues the search from this augmented subgoal node. This algorithm, called *subgoal augmentation*, is summarized in Figure 2 and Algorithm 1. We iteratively run this algorithm until all subgoal nodes have been incorporated into the search trace. An example of the resulting search trace is provided in Appendix C. Finally, the model is fine-tuned on the successful search traces via supervised learning over multiple iterations. Algorithm 2 summarizes the overall training procedure.

A key advantage of Guided-ReST is its ability to explicitly teach the model where to backtrack during unsuccessful searches. Unlike ReST, which treats failed search traces as uninformative, or RL, which relies solely on terminal reward signals, it leverages optimal subgoal information to provide corrective feedback at failure points and guide the model to continue the search from more promising points. By systematically injecting the subgoal information to search traces, we expose the model to high-quality local corrections grounded in successful reasoning steps. This enables the model not only to generate correct answers but also to internalize structured reasoning strategies and the ability to recover from failure.

### 3.3 Operation-level RL fine-tuning

Building upon the fine-tuned model using Guided-ReST, we further improve its performance via RL fine-tuning using PPO. In standard PPO fine-tuning, the importance ratio is computed in a token-level Markov Decision Process (MDP). However, since our goal is to optimize the model's search strategy rather than token-level generation, we reformulate PPO in an operation-level MDP. Specifically, each action $a_h$ is defined as a sequence of tokens $(a_{h,1}, \ldots, a_{h,L})$ that represents a single tree operation. Consequently, each state $s_h$ is defined as the concatenation of the question and all previously executed operations. In this MDP formulation, the log importance ratio between the learner policy $\pi_\theta$ and the sampling policy $\pi_{\theta_{\text{old}}}$ is defined as

$$\log r_h(\theta) = \sum_{\ell=1}^{L} \left(\log \pi_\theta(a_{h,\ell} \mid a_{h,1:\ell-1}, s_h) - \log \pi_{\theta_{\text{old}}}(a_{h,\ell} \mid a_{h,1:\ell-1}, s_h)\right)$$

---

**Algorithm 3** Guided reinforced self-training (episode-level)

---

**Require:** model $\pi_{\text{ref}}$, question $q$, optimal solution $S$, maximum turn limit $T$

1: Initialize model parameters: $\pi_\theta \leftarrow \pi_{\text{ref}}$
2: **for** $i = 1, 2, \ldots, \text{MAXITER}$ **do**
3:     Generate a multi-turn episode: $Z \leftarrow \pi_\theta(\cdot \mid q)$
4:     **for** $t = 1, 2, \ldots, T$ **do**
5:         **if** $Z$ is successful **then**
6:             **break**
7:         **end if**
8:         *# simplified subgoal augmentation*
9:         Truncate the episode up to the $t$-th turn: $Z \leftarrow (S_1, E_2, \ldots, E_t, S_t)$
10:         Append user feedback containing the optimal solution $Z \leftarrow (S_1, E_2, \ldots, E_t, S_t, E'_{t+1})$
11:         Continue the episode: $Z \leftarrow \pi_\theta(\cdot \mid q, Z)$
12:     **end for**
13:     Fine-tune the model via supervised learning: $\pi_\theta \leftarrow \text{TRAIN}(\pi_{\text{ref}}, q, Z)$
14: **end for**

---

Finally, we train the learner policy $\pi_\theta$ and the value function $V_\phi$ with the standard PPO objective:

$$\max_\theta \ \mathbb{E}_{(s_h, a_h) \sim \pi_{\theta_{\text{old}}}} \left[ \min \left( r_h(\theta) A_h, \text{clip} \left( r_h(\theta), 1 - \epsilon, 1 + \epsilon \right) A_h \right) \right],$$

$$\min_\phi \ \mathbb{E}_{(s_h, a_h) \sim \pi_{\theta_{\text{old}}}} \left[ \tfrac{1}{2} \left( V_\phi(s_h) - R_h \right)^2 \right],$$

where $\epsilon > 0$ is the clipping coefficient. We compute the advantage $A_h$ and return $R_t$ using the Monte Carlo method instead of GAE [24], and remove the KL penalty term, following the recent practice of Yu et al. [40].

## 4 Application to episode-level search: code self-repair

So far, we have established our method on Countdown, a challenging yet formally defined task with a finite search space. In this section, we extend it to more realistic domains involving longer reasoning chains and more complex responses. Unlike Countdown, where the entire step-wise search trace fits within the context window, the increased output length and complexity make fine-grained tree search impractical. Therefore, we consider episode-level search, which generates complete solution attempts and iteratively refines them through multiple rounds of revision [20, 29, 10]. Specifically, given a question $q$, a search trace $Z$ is represented as a multi-turn episode:

$$Z = (S_1, E_2, \ldots, E_T, S_T),$$

where $S_t$ denotes a complete set of reasoning steps and the resulting solution generated by the model and $E_t$ denotes user feedback containing verification results for the previous model's response and a revision instruction. The episode ends when the model's response passes verification or the maximum turn limit $T_{\max}$ is reached.

In this setting, we apply our method to the code self-repair task, where the model first generates an initial code solution and then iteratively refines it based on verification results from a public test set. The optimal solution is relatively easy to obtain because correctness can be automatically verified via code execution, making this task an effective testbed for evaluating our method. However, directly applying our method to this task still presents challenges, since subgoals are difficult to define within either the optimal solution or the search trace.

To address this issue, we employ a simplified, episode-level variant of Guided-ReST that progressively guides the model through user feedback augmented with the optimal solution as a hint. At each round $t$, the algorithm first truncates the current unsuccessful episode up to the $t$-th turn, then augments the subsequent user feedback $E'_{t+1}$ with the optimal solution, and finally regenerates the remaining turns to complete the episode. The process repeats until the round reaches the maximum turn limit. As the round progresses, the number of turns augmented with the optimal-solution hint gradually increases, yielding progressively guided search traces. Finally, the model is fine-tuned on the successful search traces via supervised learning over multiple iterations. Algorithm 3 summarizes the overall training procedure. Additional details are provided in Appendix D.

# 5 Experiments

## 5.1 Setup

### 5.1.1 Countdown

We use Llama-3.2-1B-Instruct as the base model [5], with a maximum response length of 4K tokens.[2] For SoS, we generate search traces using heuristic-guided DFS and BFS over 500K training examples and fine-tune the base model for two epochs. For Guided-ReST, we generate a single search trace for each problem using 200K training examples and fine-tune the model for two epochs, repeating this procedure for three iterations. For PPO, we fine-tune the model on 200K training examples for two epochs. We adopt an outcome reward function that assigns 1 for success and 0 otherwise. Additional details are provided in Appendix E.

For evaluation, we follow the protocol of Gandhi et al. [4]. We construct 10K test examples for each of two settings: (1) seen targets, where the target numbers overlap with those in the training data but are paired with different input numbers, and (2) unseen targets, where the target numbers are entirely disjoint from the training data. We measure accuracy while varying the token budget.

We compare our method against BC, SoS, and the two fine-tuning methods introduced in Gandhi et al. [4]: (1) ReST and (2) PPO. Although the original SoS paper employs iterative advantage-induced policy alignment (APA) as the base RL algorithm [43], we find that PPO outperforms iterative APA. Unless otherwise specified, we conduct all RL fine-tuning experiments using operation-level PPO.

### 5.1.2 Code self-repair

We use Qwen2.5-7B-Instruct as the base model [36], with a maximum response length of 16K tokens. We build the training data using PrimeIntellect's SYNTHETIC-1 [16], which was curated from APPS, CodeContests, and TACO [8, 14, 13]. We select problems that have at least one ground-truth solution, resulting in 16K problems. For Guided-ReST, We generate eight episodes per problem, each with a maximum of four turns, and fine-tune the model for two epochs, repeating this procedure for three iterations. We compute the loss only on the last model response, following the practice of Snell et al. [29]. Additional details are provided in Appendix E.

For evaluation, we consider two code benchmarks: (1) CodeContests and (2) CodeForces [14, 21]. CodeContests consists of 165 problems and represents a near-distribution evaluation, as our training data was partially curated from the same source. On the other hand, CodeForces consists of 408 more recent problems, providing a more challenging out-of-distribution benchmark. We measure accuracy by extracting code from the model's response and executing it on private test cases.

We compare our method against the base model and ReST. We do not perform RL fine-tuning due to limited computational resources. Instead, we evaluate performance using pass@$k$ accuracy, which is a reliable proxy for RL fine-tuning [41].

## 5.2 Countdown results

Figure 3 shows the performance of each method across different token budgets on Countdown, where responses are sampled using greedy decoding. Our method significantly outperforms SoS and the baseline fine-tuning methods. At the maximum token budget, our method achieves 87% accuracy on both seen and unseen targets, outperforming the second-best PPO by more than 10%. It is important to note that the improved performance on unseen targets indicates that our method does not merely memorize the optimal solutions provided as guidance. Instead, it internalizes a more effective search strategy, leading to stronger generalization. Moreover, our method achieves comparable accuracy to the second-best PPO while consuming only about 50% of its tokens, demonstrating a more efficient utilization of test-time compute. BC achieves less than 40% accuracy on both seen and unseen targets, indicating substantially weaker generalization performance compared to search-based models.

---

[2]We choose the Llama-3 model for Countdown due to its efficient tokenization of integers.

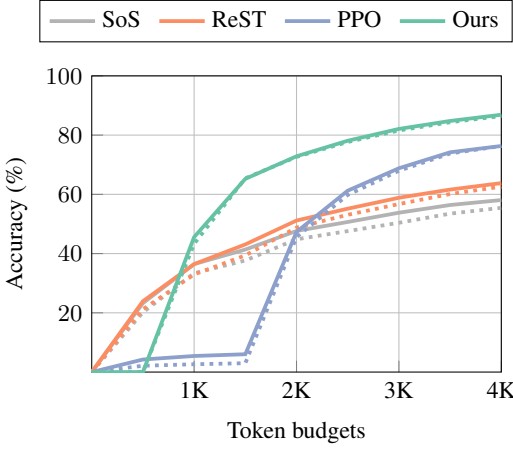
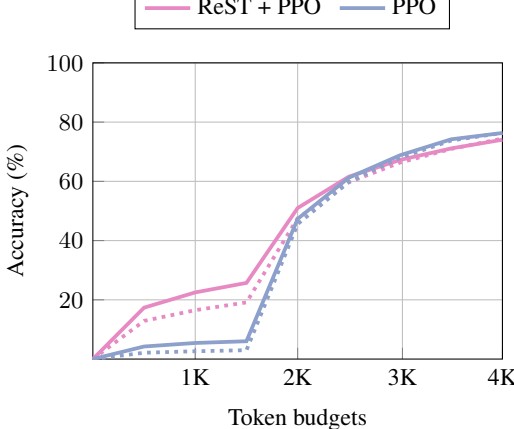

Figure 3: Accuracy of our method and baselines on Countdown. Solid lines represent seen targets, and dotted lines represent unseen targets.

Figure 4: Accuracy of PPO initialized with ReST on Countdown. PPO does not benefit from ReST in the high-budget regime.

Table 2: Pass@$k$ accuracy of ReST and Guided-ReST on Countdown (32 samples per problem).

| Method | Seen target | | | | | | Unseen target | | | | | |
|---|---|---|---|---|---|---|---|---|---|---|---|---|
| | 1 | 2 | 4 | 8 | 16 | 32 | 1 | 2 | 4 | 8 | 16 | 32 |
| SoS | 55.3 | 63.8 | 69.6 | 73.3 | 75.5 | 77.1 | 53.2 | 62.3 | 68.7 | 73.0 | 75.6 | 77.5 |
| ReST | 62.3 | 67.7 | 71.3 | 73.6 | 75.3 | 76.6 | 60.8 | 66.7 | 70.8 | 73.5 | 75.5 | 77.0 |
| Guided-ReST | **62.7** | **75.3** | **84.6** | **90.8** | **94.7** | **96.8** | **61.0** | **74.1** | **83.8** | **90.3** | **94.3** | **96.8** |

## 5.3 Impact of Guided-ReST on RL fine-tuning

Our method leverages both Guided-ReST and PPO for fine-tuning. This naturally raises the question of whether PPO can achieve similar synergy with standard ReST, which does not leverage optimal-solution guidance during data generation. Figure 4 presents the accuracy of PPO fine-tuning initialized with ReST. We observe that PPO does not benefit from ReST, but achieves significant improvements when combined with Guided-ReST.

To examine why Guided-ReST exhibits stronger synergy with PPO, we evaluate the pass@$k$ accuracy. Table 2 presents the pass@$k$ accuracy of ReST and Guided-ReST on Countdown, where responses are sampled at a temperature of 1.0. ReST outperforms SoS at pass@1 but shows almost no difference at pass@32. Recent work suggests that RL mainly shifts probability mass from already-likely correct responses toward top-ranked ones [41]. Consequently, since ReST and SoS exhibit similar coverage of correct candidates, PPO initialized with ReST yields performance comparable to that initialized with SoS. In contrast, Guided-ReST achieves a similar pass@1 accuracy to ReST but delivers much larger accuracy gains as $k$ increases. This broader coverage provides PPO with more correct candidates to amplify, leading to significant performance improvements.

## 5.4 Analysis of operation-level MDP

We investigate the effectiveness of the operation-level MDP formulation for RL fine-tuning introduced in Section 3.3. Figure 5 shows the performance of our method under the standard token-level MDP. We find that fine-tuning the model under the operation-level MDP not only improves performance from 83% to 87% at the maximum token budget but also substantially enhances test-time compute efficiency. Compared to the token-level MDP, our approach achieves approximately 2× higher token efficiency in the low-budget regime and about 1.5× higher efficiency in the high-budget regime. This result highlights the importance of defining an MDP aligned with the optimization objective.

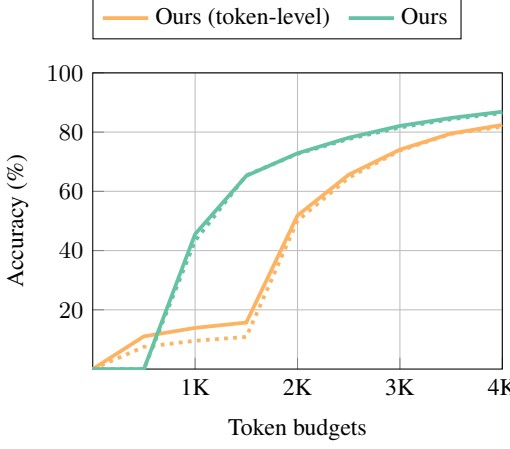

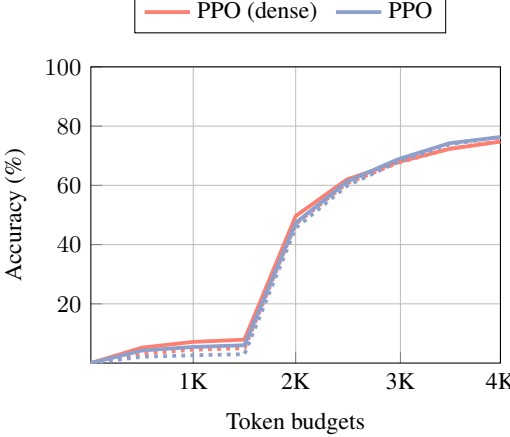

Figure 5: Accuracy of our method when trained under the token-level MDP on Countdown.

Figure 6: Accuracy of PPO trained with a dense reward on Countdown.

Table 3: Pass@$k$ accuracy of ReST and Guided-ReST on code self-repair (128 samples per problem).

| Method | CodeContests | | | | | | CodeForces | | | | | |
|---|---|---|---|---|---|---|---|---|---|---|---|---|
| | 1 | 2 | 4 | 8 | 16 | 32 | 1 | 2 | 4 | 8 | 16 | 32 |
| Base | 4.5 | 7.5 | 11.3 | 15.7 | 20.6 | 25.8 | 5.5 | 8.5 | 12.5 | 17.2 | 22.4 | 27.7 |
| ReST | 9.4 | 13.1 | 17.0 | 21.3 | 25.9 | 30.4 | **9.7** | 14.2 | 19.3 | 24.8 | 30.5 | 35.9 |
| Guided-ReST | **10.5** | **14.8** | **19.4** | **24.1** | **28.9** | **33.9** | **9.7** | **14.5** | **20.2** | **26.2** | **32.0** | **37.6** |

## 5.5 RL fine-tuning with dense reward

Our method leverages subgoal guidance derived from optimal solutions during data generation. As an alternative, one can directly perform RL fine-tuning using subgoal-based rewards. Given an optimal solution $S = (s_1, \ldots, s_T)$ and a search trace $Z = (z_1, \ldots, z_{T'})$, we define the subgoal reward as

$$R_{\text{subgoal}}(z_{1:t'} \mid q) = \begin{cases} 0.25 & \text{if there exists } s_t \text{ such that } z_{t'} \text{ reaches } s_t \\ 0 & \text{otherwise,} \end{cases}$$

We fine-tune the SoS model using PPO with a dense reward that combines the subgoal and outcome reward functions.[3] To prevent reward hacking, the subgoal reward is applied only once per subgoal.

Figure 6 presents the performance of PPO with the dense reward. The dense reward has little effect on overall performance and even causes a slight degradation in the high-budget regime. These results underscores the importance of Guided-ReST in conditioning the model before the RL phase.

## 5.6 Code self-repair results

As observed in the Countdown experiment in Section 5.3, Guided-ReST achieves similar pass@1 accuracy to ReST but yields higher pass@$k$ as $k$ increases, which translates into greater improvements during PPO fine-tuning. To examine whether this effect generalizes beyond Countdown, we conducted the same analysis on the code self-repair task.

Table 3 presents the pass@$k$ accuracy of ReST and Guided-ReST on CodeContests and CodeForces. Guided-ReST consistently achieves higher accuracy than ReST, with the gap between them widening as $k$ increases. Note that ReST also outperforms the base model, as the base model is general-purpose rather than task-specific. The overall improvement remains moderate compared to Countdown, likely due to fewer training examples, incomplete subgoal augmentation, and weaker search capabilities.

---

[3]The total reward ranges from 0 to 1.5, as each Countdown problem contains two non-trivial subgoals.

## 6 Related work

**Searching with language models**    Yang et al. [37] first introduce the concept of training a sequence model to imitate expert MCTS policy search traces, primarily to enhance imitation learning. More recent studies have shifted focus toward directly improving search capabilities by training language models on these traces and fine-tuning them with self-generated data [11, 4]. In contrast to these prior studies, which apply generic self-improvement algorithms not tailored for search [42, 6], our work analyzes the relationship between the quality of self-generated data and the efficiency of test-time compute, and introduces a self-improvement algorithm specifically optimized for search efficiency. Several studies have also explored sequential revision and episode-level search [35, 20, 29], but these do not address improvements in test-time compute efficiency.

**Fine-tuning on self-generated data**    Anthony et al. [2] first introduce the idea of iteratively distilling self-generated data into neural networks, generating high-quality trajectories using an expert MCTS policy and imitating them to improve a neural network-based policy. Recent studies have extend this idea to fine-tune language models on self-generated data for various tasks, including mathematical reasoning, question answering, and machine translation [19, 42, 6], although these approaches are not specifically tailored for search. The work most closely related to ours is that of Chang et al. [3], which employs a stronger guide model alongside a weaker base model to generate data and fine-tune the base model using PPO. Their method, however, is limited to a single round of interaction between the guide and base models. Our approach performs multiple rounds of guide-base interactions, enabling richer and more iterative supervision.

## 7 Conclusion

We introduce Guided-ResT, a self-training algorithm designed to improve the search efficiency of language models. By incorporating optimal solution as guidance for generating high-quality search traces, our method enables models to internalize more effective search strategies. Experiments on the Countdown and code self-repair tasks demonstrate that our method not only enhances accuracy but also scales robustly with increased test-time compute, outperforming existing fine-tuning approaches.

Our work is not without limitations. It assumes access to optimal solutions, which may not always be available in practice. Nevertheless, this assumption can be relaxed by leveraging solutions generated by a sufficiently capable model. Once such a model is available, a promising direction is to develop a collaborative framework in which a stronger model provides guidance when the base model fails to make progress, and this interaction is subsequently distilled into the base model through additional training. This teacher-student framework facilitates the transfer of effective search strategies from the teacher to the student model.

Another limitation lies in the assumption of an oracle in Countdown that can precisely identify the first incorrect step and backtrack to that point. Although such an oracle provides a clear and reliable signal for targeted correction, this assumption becomes unrealistic when replaced with a language model, particularly standard instruction-following models that still struggle to recognize reasoning errors or perform localized backtracking. A promising direction for future work is to extend our approach to reasoning models that already exhibit emerging backtracking and self-correction capabilities [7].

## Acknowledgements

This work was supported in part by Institute of Information & Communications Technology Planning & Evaluation (IITP) grant funded by the Korea government (MSIT) (No. RS2020-II200882, (SW STAR LAB) Development of deployable learning intelligence via self-sustainable and trustworthy machine learning, No. RS-2022-II220480, Development of Training and Inference Methods for Goal-Oriented Artificial Intelligence Agents, No. RS-2021-II211343, Artificial Intelligence Graduate School Program (Seoul National University)), and by the National Research Foundation of Korea (NRF) grant funded by the Korea government (MSIT) (No. RS-2024-00354036). This research was supported by a grant from KRAFTON AI. This material is based upon work supported by the Air Force Office of Scientific Research under award number FA2386-23-1-4047. Hyun Oh Song is the corresponding author.

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
