# OpenReview forum: "Learning to Better Search with Language Models via Guided Reinforced Self-Training"
_NeurIPS.cc/2025/Conference — NeurIPS 2025 poster_

### Official Review · Reviewer_sgBD · 2025-06-28

**Clarity:** 2
**Significance:** 2
**Originality:** 2
**Rating:** 3
**Confidence:** 3

**Summary:**

This work proposes Guided-ReST that integrates optimal solutions into the model’s search procedure. It leverages unsuccessful search attempts as context for each intermediate reasoning step. Then, Guided-ReST conducts finetuning on these automatically generated search traces. The experiments are evaluated on Countdown, MATH-500, and AMC23, where Guided-ReST shows improvements over baselines in terms of the same token budget.

**Questions:**

1. How do we define nodes for other reasoning tasks like multi-hop reasoning or coding?

**Ethical Concerns:**

["NO or VERY MINOR ethics concerns only"]

**Final Justification:**

I thank the rebuttal from the authors. I appreciate the results from llama and observe its comparable performance with the GPT-2 model. Though the author provides justifications on how the proposed method generalize to other task domains, the evaluation from the original paper is still only conducted in a limited domain with a particular focus on the Countdown tasks. Given the factors above, I have increased my final score to 3.

**Limitations:**

Yes.

**Paper Formatting Concerns:**

No.

**Quality:**

2

**Strengths And Weaknesses:**

Strength:

1. Incorporating the partial solution as an initial prompt to guide the model to generate more diverse and effective reasoning trajectories is a natural idea and effective in the data preparation stage.

2.  Guided-ReST shows improved performance on the Countdown task with a good coverage of baselines, such as BC, SoS, ReST, and PPO

Weakness:

1. My major concern for this work is about the choice of the backbone models and the benchmarks. First, there is an inconsistent choice of base models without enough justification used in the Countdown task - GPT2-250M and mathematical reasoning tasks Llama-3.1-8B-it, respectively. Since the method is mainly developed for the Countdown task, the 250M model may not be evident enough to reflect the generalization of the model to other scales of the base model.

2. The method itself, as acknowledged by the author in the limitation section, requires optimal step-by-step solutions, and the reasoning steps need to be easily decomposed into steps. However, the evaluation on Countdown and two mathematical reasoning tasks is not convincing to demonstrate the effectiveness of the method to other reasoning tasks. I suggest that authors include more base models and benchmarks.

---

> ### Author Rebuttal · Authors · 2025-07-30
>
> Thank you for your helpful feedback, particularly regarding the choice of backbone models. We would like to address your questions below.
>
> ---
>
> **Q1. The choice of the backbone models**
>
> Thank you for raising this point. Following the protocol in the stream-of-search (SoS) paper [1], our Countdown experiments used a GPT-2 model with 250M parameters trained from scratch. While this gave us a clean baseline–a fully controlled model trained from scratch–we recognize that modern LLMs differ greatly from GPT-2 in scale, architecture, and tokenization. Consequently, we acknowledge your concern that findings based on a GPT-2 model trained from scratch may not readily generalize to modern LLMs.
>
> To check how well our approach generalizes, we repeated the Countdown experiments with a more recent LLM. Because our paper already employs Llama-3.1-8B-Instruct for the mathematical reasoning experiments, we chose Llama-3.2-1B-Instruct–a model from the same Llama-3 family–for the Countdown task. We conducted the experiments with the same hyper-parameters as in the original setup, but reduced the number of epochs to 2. In addition, we reformatted the dataset into an instruction-chat style. The results are summarized below, showing that our method scales to larger, instruction-tuned backbones.
>
> | Method | Accuracy (seen) | Accuracy (unseen) |
> | --- | --- | --- |
> | SoS (GPT-2) | 0.5747 | 0.5342 |
> | SoS (Llama) | 0.5718 | 0.5391 |
> | ReST (GPT-2) | 0.6279 | 0.6003 |
> | ReST (Llama) | 0.6330 | 0.6033 |
> | Guided-ReST (GPT-2) | 0.6823 | 0.6698 |
> | Guided-ReST (Llama) | 0.6791 | 0.6659 |
> | Guided-ReST+PPO* (GPT-2) | 0.7507 | 0.7361 |
> | Guided-ReST+PPO* (Llama) | 0.7561 | 0.7420 |
> (* denotes the method presented as Ours in the paper.)
>
> ---
>
> **Q2. More benchmarks**
>
> Regarding a new benchmark, We are exploring an additional experiment that applies our method to a code-editing setting. While we may not have enough time to complete this study within the rebuttal period, we will run it regardless to strengthen our paper. If the results are ready before the rebuttal period ends, we will share them with you.
>
> ---
>
> **Q3. Definition of nodes in multi-hop reasoning or coding**
>
> We provide a concrete example of nodes in multi-hop reasoning tasks, using problems from PrOntoQA [2] and MuSiQue [3], which are well-known benchmarks for multi-hop logical reasoning and question answering.
>
> **PrOntoQA**
>
> Context:
> - Each cat is a carnivore.
> - Every carnivore is not herbivorous.
> - Carnivores are mammals.
> - All mammals are warm-blooded.
> - Mammals are vertebrates.
> - Every vertebrate is an animal.
> - Animals are multicellular.
>
> Question: Fae is a cat. True or false: Fae is not herbivorous
>
> Solution:
> 1. Fae is a cat.
> 2. Cats are carnivores.
> 3. Fae is a carnivore.
> 4. Every carnivore is not herbivorous.
> 5. Fae is not herbivorous
>
> In this case, each reasoning step forms a node in the search trace. These nodes are connected sequentially, with each step building on the previous one, forming a sequential reasoning graph (1 &rarr; 2 &rarr; 3 &rarr; 4 &rarr; 5).
>
> **MuSiQue**
>
> Question: When did the people who captured Malakoff come to the region where Philipsburg is located?
>
> Solution:
> 1. What is Philipsburg capital of? Saint Martin
> 2. Saint Martin is located on what terrain feature? Caribbean
> 3. Who captured Malakoff? French
> 4. When did the French come to the Caribbean? 1625
>
> Coding problems like MBPP [4] can also be framed as search problems, where the goal is to find an optimal solution (code) that satisfies the given prompt. In this context, search traces would represent the process of iterating over potential solutions by generating a new code snippet or modifying an existing one.
>
> ---
>
> Again, we sincerely appreciate your constructive feedback, which has been very helpful in improving our paper. We hope that our response above addresses all of your questions.
>
> ---
>
> **References**
>
> [1] Kanishk Gandhi et al., Stream of Search (SoS): Learning to Search in Language, COLM 2024
> [2] Abulhair Saparov and He He, Language Models Are Greedy Reasoners: A Systematic Formal Analysis of Chain-of-Thought, ICLR 2023
> [3] Harsh Trivedi et al., MuSiQue: Multihop Questions via Single-hop Question Composition, arXiv 2021 \
> [4] Jacob Austin et al., Program Synthesis with Large Language Models, arXiv 2021

---

### Official Review · Reviewer_yPG3 · 2025-07-02

**Clarity:** 2
**Significance:** 3
**Originality:** 2
**Rating:** 5
**Confidence:** 2

**Summary:**

This paper proposes a fine-tuning algorithm for LLMs. Instead of using complete thought traces (as done in the Stream of Search paper), the approach distills effective search traces, used later for fine-tuning. These optimal search solutions are generated using guided subgoal augmentation that replaces random unsuccessful search traces with new searches.
The results show how this technique improves LLMs search capabilities for arithmetic and mathematical reasoning tasks.

**Questions:**

- In Figure 2, the guided subgoal augmentation algorithm is shown. Why is the right path (the one that has node 9 and then 10) removed at step 2? The selected node (at step 1) is 4, and correctly, all the history coming from that is deleted. I do not understand why the path with nodes 9 and 10 is also removed, since it stems from node 1, not node 4.
- What is the reasoning behind choosing  Llama-3.3 70B-Instruct for generating the step-by-step solutions (Section 5.1.2)?
- Are the results in Figures 4-5-6-7 representing the performances on Countdown? This is not outlined either in the caption or when referencing those figures.

**Ethical Concerns:**

["NO or VERY MINOR ethics concerns only"]

**Final Justification:**

The authors clearly answered questions and argued the significance of this work. Various improvements have been made, including the addition of computational overhead metrics, metric variance measurements, and experiments using llama models. This raises the quality of this work. Due to this, I raise my score to Accept.

**Limitations:**

While the limitations are outlined in the Conclusion section, they could be more detailed. The proposed solution is not sufficient to address the shortcoming previously explained. The criteria to distinguish 'standard' and 'long' CoT models should be outlined earlier in the paper.

**Paper Formatting Concerns:**

No concerns

**Quality:**

3

**Strengths And Weaknesses:**

Strengths:
- The paper communicates the main ideas clearly.
- The references are plentiful and proper, and the authors provide clear attribution when borrowing ideas from other papers.
- The claims are supported by the results.

Weaknesses:
- The approach for generating training data relies on having ground truth solutions, which may not be available for the most challenging reasoning tasks;
- Experiments on the BigMath dataset (Section 5.1.2) rely on Llama 3.3 70B-generated solutions as substitutes for missing optimal solutions, but there's no verification that these generated solutions are actually optimal, which undermines the quality of the training data.
- The paper is mostly clearly written, but the Experiments section could be restructured. I also suggest having a separate section for results to improve readability.
- Lines 20-21: "Their impressive performance is primarily attributed to auto-regressive training on high-quality, internet-scale data". While internet-scale data is correct, claiming that the data used for training is high-quality is debatable, considering that for closed-source models we do not know exactly what data was used for training.

---

> ### Author Rebuttal · Authors · 2025-07-30
>
> Thank you for the valuable feedback, which has greatly helped us improve the clarity of our paper. We would like to address your questions below.
>
> ---
>
> **Q1. Reliance on ground-truth solutions**
>
> We acknowledge that relying on exact optimal traces limits the immediate applicability of our method to general reasoning tasks. For that reason, we use the term "search" rather than "reasoning" in the title and throughout the main text, reserving "reasoning" only for fixed phrases such as "reasoning step," "arithmetic reasoning," and "mathematical reasoning," where it is standard and aids clarity. Although our method rests on a strong assumption, we believe it nevertheless offers valuable insights into two fundamental questions.
>
> 1. Can outcome-level rewards alone improve an LLM's in-context search capabilities?
> 2. If outcome rewards are insufficient, what additional supervision is needed?
>
> Our results indicate that neither ReST nor PPO alone uncovers new search capabilities. The model develops such capabilities only when step-by-step guidance is added through supervised learning. To substantiate this claim, we report pass@k accuracy below–a metric widely used in recent work to test whether a model truly acquires novel reasoning abilities [1, 2].
>
> | Method | Pass@1 | Pass@2 | Pass@4 | Pass@8 | Pass@16 | Pass@32 | Pass@64 |
> | --- | --- | --- | --- | --- | --- | --- | --- |
> | SoS | 0.551 | 0.633 | 0.686 | 0.736 | 0.774 | 0.805 | 0.841 |
> | ReST | 0.601 | 0.652 | 0.700 | 0.732 | 0.763 | 0.788 | 0.809 |
> | Guided-ReST | 0.630 | 0.731 | 0.835 | **0.896** | **0.944** | **0.965** | **0.975** |
> | Guided-ReST+PPO* | **0.734** | **0.793** | **0.839** | 0.883 | 0.903 | 0.925 | 0.941 |
> (* denotes the method presented as Ours in the paper.)
>
> While ReST surpasses SoS at pass@1, its accuracy slips below SoS from pass@8 onward, suggesting that ReST mainly boosts the probability of solutions already favored by the model rather than uncovering new ones. By contrast, Guided-ReST improves accuracy at every k, indicating that it enables the model to develop genuinely novel search capabilities. We will state our contributions explicitly in the revised manuscript.
>
> In addition, we will add a dedicated “Limitations” section. The main limitations we have identified when applying our method to general-purpose reasoning tasks are as follows:
> 1. Token efficiency
>    - Countdown involves only three hops, and each CoT step is a single arithmetic operation, so the entire search trace remains short and easy to log (still it consumes 4096 tokens to achieve strong performance).
>    - In typical reasoning tasks, the depth is much greater and each step can be multiple sentences, so the entire search trace grows rapidly.
> 2. Branching factor
>    - In Countdown, the branching factor is fixed and relatively small.
>    - In natural language reasoning tasks, the branching factor is unbounded.
> 3. Existence of oracle
>    - In Countdown, we assume an oracle that pinpoints the first incorrect step in the search track, backtracks to it, and generate the correct next reasoning step.
>    - In general reasoning tasks, these verification and generation steps can be noisy.
>
> ---
>
> **Q2. Use of Llama-3.3-70B-Instruct**
>
> We followed the same protocol as the Big-Math paper [3]–generating solutions with Llama-3 models and validating them using the final answers–but opted for Llama-3.3-70B-Instruct instead of Llama-3.1-405B-Instruct. The 70B model fits on a single node with 8 GPUs without quantization, yet reportedly matches (and on the MATH benchmark even surpasses) the performance of the 405B model. In this work, we measure solution quality only by correctness, without evaluating clarity, succinctness, or other factors. Therefore, we acknowledge that, Llama-3.3-70B-Instruct may not always produce the highest-quality answers on dimensions beyond correctness, and can sometimes produce the correct final answer even though the reasoning it provides is wrong.
>
> ---
>
> **Q3. Clarity of the experiment section**
>
> Thank you for the suggestion. In the revised manuscript, we will add separate sections for the Countdown experiments and the math-reasoning experiments.
>
> ---
>
> **Q4. Line 20-21**
>
> Thank you for pointing it out. We will correct the sentence to: “auto-regressive training on massive web-scale data.”
>
> ---
>
> **Q5. Figure 2**
>
> You are correct that, in an abstract search tree, modifying node 4 does not change the branch that contains node 9 and 10. However, in our LLM-driven search, the model explores nodes sequentially and every new expansion is conditioned on the full context of all nodes visited so far. Concretely, when the model reached node 9, its prompt already included the original contents of node 4. If we later revise node 4, then the prompt becomes stale. Therefore, we discard all nodes visited after the modified node–even those on different logical branches–because they were generated from an outdated context. We will clarify this in the revised manuscript.
>
> ---
>
> **Q6. Figures 4-5-6-7**
>
> Yes, these figures report the performance on the Countdown task. We will make this explicit in each caption.
>
> ---
>
> We again thank you for providing constructive feedback, which truly enhances the quality of our paper. We hope that our response adequately addresses your questions.
>
> ---
>
> **References**
>
> [1] Yang Yue et al., Does Reinforcement Learning Really Incentivize Reasoning, arXiv 2025 \
> [2] Andre He et al., Rewarding the Unlikely: Lifting GRPO Beyond Distribution Sharpening, arXiv 2025 \
> [3] Alon Albalak et al., Big-Math: A Large-Scale, High-Quality Math Dataset for Reinforcement Learning in Language Models, arXiv 2025

---

> > ### Comment · Reviewer_yPG3 · 2025-08-05
> >
> > Thank you to the authors for clearly addressing a variety of reviewers' questions. I also appreciate the addition of results using the llama models and the addition of confidence intervals. Due to this, i raise my score to Accept.

---

> > > ### Author Response · Authors · 2025-08-08
> > >
> > > We appreciate your acknowledgment of the additional results and clarifications we have provided. We will definitely include the Llama experiments in the main text to strengthen our claim and add a separate section detailing the limitations of our method. We sincerely thank you once again for dedicating your time to the review and rebuttal process.

---

### Official Review · Reviewer_5NRh · 2025-07-03

**Clarity:** 2
**Significance:** 3
**Originality:** 3
**Rating:** 4
**Confidence:** 3

**Summary:**

The paper introduces a new fine-tuning method called Guided-ReST to improve the search abilities of language models. The method trains the model to "correct" its own search path/tree, trained on a synthetic dataset generated using optimal solutions. The paper shows this method ourperforms baseline fine-tuning methods on Countdown and math datasets.

**Questions:**

Please focus on addressing aforementioned weaknesses. I will consider raising my score if the following questions are well-addressed:
1. How does the method extend to general reasoning tasks? What are the limitations? Discussion points may include:
  a) Does search tree / search path make a difference?
  b) The paper uses a stronger model as a substitue for the optimal solution in mathematical reasoning. Is the performance then bottlenecked by the strong model?
2. Can you clarify if the main contributions are limited to a data generation scheme during training?

I will most likely not lower my score as long as the following questions are addressed:
1. Can you propose some revision to Section 3 to improve readability? Specifically, the main confusion may result from how $Z$ is framed in Algorithms 1 and 2. My understanding is that the revised search path is appended to the text during training to obtain a long CoT, but this isn't clearly stated.
2. Can you discuss some other fine-tuning methods, and why they are not included as baselines?

**Ethical Concerns:**

["NO or VERY MINOR ethics concerns only"]

**Final Justification:**

The authors have addressed my concerns through the rebuttal, and now I believe this work is a good contribution to the field. Despite this, I raised the score to borderline accept since I think the contributions are not incredibly significant, but it's a good work.

**Limitations:**

Yes, though I would appreciate a more in-depth discussion on limitations and how the methodologies can be applied to complex CoT models, perhaps in the appendix.

**Quality:**

3

**Strengths And Weaknesses:**

Strengths:
The method is well-motivated, the introduction is intuitive, and the conclusions are clear. The experiment results are well-presented. The idea of guided search, a form of refinement, is quite interesting.

Weaknesses:
1. The paper mainly illustrates the method on Countdown, where the reasoning problem is easily represented as a search problem. While the authors also demonstrate the method's effectiveness on math, relevant paragraphs are scarce, and discussions on how the method extends to general reasoning problems seem to be lacking.
2. The main contribution seems to be limited to a data generation scheme in SFT, where an optimal search path is utilized to help the model more accurately pinpoint the current search bottleneck.
3. Some parts of section 3 is confusing to me. For one, it is not fully clear what model inference looks like. According to the appendices, the search trace has crucial steps "Moving to Node #..." for Countdown and "There may be an error in your solution" prompt for math, characterizing the LLM's self-adjustment. I feel like these should also be discussed briefly in the main paper.

---

> ### Author Rebuttal · Authors · 2025-07-30
>
> Thank you for your constructive feedback. It clearly highlighted which parts needed more explanation and guided us on how to improve them. We would like to address your questions below.
>
> ---
>
> **Q1. Extension to general reasoning tasks**
>
> Yes. Any task that can be solved by step-by-step chain-of-thought (CoT) reasoning can, in principle, be cast as a search problem, often called meta-CoT [1].
> - The root node is the original prompt.
> - Each node expansion appends a single reasoning step.
>
> Thus, a single root-to-leaf path in the search tree corresponds to one complete CoT.
>
> However, there might be some practical issues to apply this framework general-purpose reasoning tasks.
> 1. Token efficiency
>    - Countdown involves only three hops, and each CoT step is a single arithmetic operation, so the entire search trace remains short and easy to log (still it consumes 4096 tokens to achieve strong performance).
>    - In typical reasoning tasks, the depth is much greater and each step can be multiple sentences, so the entire search trace grows rapidly.
> 2. Branching factor
>    - In Countdown, the branching factor is fixed and relatively small.
>    - In natural language reasoning tasks, the branching factor is unbounded.
> 3. Existence of oracle
>    - In Countdown, we assume an oracle that pinpoints the first incorrect step in the search track, backtracks to it, and generate the correct next reasoning step.
>    - In general reasoning tasks, these verification and generation steps can be noisy.
>
> At a high level, our algorithm alternates between an exploratory student and an oracle teacher that provide step-level corrections. This interplay produces search traces with far more diverse backtracking scenarios than the teacher would encounter in self-play. Training on this richer distribution of errors and fixes broadens the student’s experience and can ultimately enable it to surpass the teacher.
>
> We will add a separate "Limitations and Future Work" section to the Appendix in the revised manuscript.
>
> ---
>
> **Q2. Contribution limited to data generation scheme**
>
> Thank you for raising this point. While our approach introduces a new data-generation scheme for rejection-sampling fine-tuning (RFT), its contribution goes beyond data generation alone. Specifically, we show that
> 1. Superior standalone performance (Section 5.2, Figure 3): The subgoal-augmented traces generated by our method consistently outperform the previous RFT baseline (ReST).
> 2. Strong synergy with RL fine-tuning (Section 5.3, Figure 4): When followed by PPO, our method yields large additional gains, whereas ReST shows almost no benefit from the same RL stage.
>
> These results show that the value of our work lies not simply in introducing a new data-generation procedure, but in how it enables a much more effective RFT &rarr; RL pipeline.
>
> We analyzed why our method shows a strong synergy with RL fine-tuning. Recent studies suggest that RL-based self-improvement mostly amplifies the probability of solutions that are already likely, rather than discovering entirely new ones; in other words, it improves pass@1 accuracy only if the base model already has a high pass@k accuracy for larger k [2, 3]. This observation leads to the hypothesis that our method provides a much stronger initial model with higher pass@k, giving RL more high-quality candidates to amplify.
>
> To test this hypothesis, We repeated the experiments on the Countdown task using the more recent Llama-3.2-1B-Instruct model and computed pass@k accuracy on a subset of 1,000 test samples. The results are summarized below.
>
> | Method | Pass@1 | Pass@2 | Pass@4 | Pass@8 | Pass@16 | Pass@32 | Pass@64 |
> | --- | --- | --- | --- | --- | --- | --- | --- |
> | SoS | 0.551 | 0.633 | 0.686 | 0.736 | 0.774 | 0.805 | 0.841 |
> | ReST | 0.601 | 0.652 | 0.700 | 0.732 | 0.763 | 0.788 | 0.809 |
> | Guided-ReST | 0.630 | 0.731 | 0.835 | **0.896** | **0.944** | **0.965** | **0.975** |
> | Guided-ReST+PPO* | **0.734** | **0.793** | **0.839** | 0.883 | 0.903 | 0.925 | 0.941 |
> (* denotes the method presented as Ours in the paper.)
>
> While ReST outperforms SoS at pass@1 accuracy, it falls behind SoS from pass@8 accuracy onward. Therefore, adding an RL stage after ReST may be less efficient than applying PPO directly to the SoS baseline. In contrast, our proposed data generation scheme significantly improves pass@k accuracy at every k. This implies that additional RL fine-tuning can be applied effectively.
>
> We will highlight the synergy with RL as our primary contribution in both the Introduction and Results sections of the revised manuscript.
>
> ---
>
> **Q3. Clarity of Section 3**
>
> We recognize that Section 2 does not fully explain how a search trace Z is expressed in text, which may have caused confusion when reading Section 3. A search trace is the textual log of every step the solver–symbolic or LLM–takes while traversing the search tree. Each trace consists of three operation types.
>
> 1. Expansion – creating child nodes
>    - Countdown: choose two remaining numbers and an operator; the child node is the updated set of numbers after the operation.
>    - Mathematical reasoning: generate one paragraph of reasoning; the child node is the concatenation of all previously generated paragraphs with this new paragraph.
> 2. Selection – moving to the next child node for exploration
> 3. Verification – checking a leaf node
>    - Countdown: when only one number remains, verify that it matches the target.
>    - Mathematical reasoning: when the solver produces an answer, verify that it is correct.
>
> Thus, a search trace is a sequential text record of expansion, selection, and verification lines in the exact order they occur during the search.
>
> In the Countdown task, logging a search trace is token-light–each operation fits into just two short lines. In mathematical reasoning, however, every node contains the full chain of reasoning generated so far, so expanding multiple children at once would duplicate that text for every child and quickly inflate the prompt. To keep the search trace compact, we therefore enforce the search to expand only one child at a time and move to it immediately. In this regime the combined expansion + selection step is recorded by appending just a single paragraph to the existing trace. In Section 4, we refer to this setting as “episode-level search”–as opposed to step-wise search. The following table represents how each operation is represented in text for each benchmark.
>
> | Operation | Countdown | Mathematical reasoning |
> | --- | --- | --- |
> | Expansion | "Exploring Operation...\nGenerated Node..." | A single reasoning step separated by the double newline |
> | Selection | "Moving to Node...\nCurrent State..." | None |
> | Verification | "a,b unequal: No Solution" or "a,a equal: Goal Reached" | "There may be an error in your solution..." |
>
> Returning to Section 3, Algorithm 1 details how we modify and prune an LLM-generated search trace using known sub-goal information (see also Figure 3 in the supplementary material) and then prompt the model to regenerate the trace. Algorithm 2 describes how we fine-tune the model on the data produced by Algorithm 1.
>
> In the revision, we will add a precise definition of a search trace and give a detailed explanation of how it is represented in text in Section 2.
>
> ---
>
> **Q4. Other fine-tuning methods**
>
> Our paper does not include DPO-style preference fine-tuning. Although we could have constructed preference pairs by ranking search traces on success and length, we concluded that, given a well-defined scalar reward function, directly optimizing it with PPO would be both more straightforward and more effective.
>
> ---
>
> Once again, we really appreciate your constructive feedback, which greatly help us to improve our paper. We hope that our response above addresses all of your questions.
>
> ---
>
> **References**
>
> [1] Violet Xiang et al., Towards System 2 Reasoning in LLMs: Learning How to Think With Meta Chain-of-Thought, arXiv 2025 \
> [2] Yang Yue et al., Does Reinforcement Learning Really Incentivize Reasoning, arXiv 2025 \
> [3] Andre He et al., Rewarding the Unlikely: Lifting GRPO Beyond Distribution Sharpening, arXiv 2025

---

> > ### Comment · Reviewer_5NRh · 2025-08-05
> > **Response to Rebuttal**
> >
> > I thank the authors for their thorough rebuttal. They have raised my confidence in this work, and I have raised my score.

---

> > > ### Author Response · Authors · 2025-08-05
> > >
> > > Thank you for acknowledging our rebuttal and we are glad to hear that your confidence has increased. We appreciate the time you spent discussing the paper with us, and we will be sure to incorporate the insights from our discussion into the revised manuscript.

---

### Official Review · Reviewer_xv2X · 2025-07-23

**Clarity:** 4
**Significance:** 3
**Originality:** 3
**Rating:** 5
**Confidence:** 3

**Summary:**

The paper builds on Stream-of-Search (SoS) and introduces Guided-ReST, a fine-tuning framework that improves the (inference) compute-efficiency of language-model search. Guided-ReST treats each optimal solution as a sequence of landmark sub-goals: during self-generation it replaces a failing child node in the search tree with the next correct sub-goal, then lets the model resume its own exploration, creating high-likelihood, high-quality traces. These augmented traces are distilled back into the model through several supervised iterations, followed by RL that further refines the policy while shortening the credit-assignment horizon.  Experiments rely on optimal solutions with subgoals and provide mixed results on the arithmetic Countdown benchmark and is also adapted with operation-level MDP to MATH-500 and AMC23.

**Questions:**

- Could you provide approximate training GPU-hours/FLOPs and average inference latency (ms per token or ms per problem) for Guided-ReST versus baselines? This would let readers verify that the method is compute-efficient in wall-clock terms, not just by token budget.
- Could you please provide 95% confidence intervals to establish that the gain is statistically reliable?
- Do you have quantitative data—e.g., distributions of log-likelihood, median trace length, or success-per-1k tokens—showing that sub-goal augmentation yields measurably higher-quality traces and that this correlates with accuracy gains?
- I’m not sure how manageable it would be but can you run Guided-ReST on at least one contemporary 7–13B open model on a different task that is not saturated easily by such large models to establish that the observed gains (with confidence intervals) are not limited to the 250M-parameter regime? This would solidify the validity of the main claim.

**Ethical Concerns:**

["NO or VERY MINOR ethics concerns only"]

**Final Justification:**

The author's rebuttals addressed most of my concerns. I have therefore increased my rating.

**Limitations:**

Training compute overhead and overall latency/FLOPs is an aspect that has not been discussed. Scalability beyond small-scale models is questionable. Reliance on optimal solutions is acknowledged by the authors.

**Quality:**

3

**Strengths And Weaknesses:**

Strengths:

- The authors clearly articulate the problem, noisy search traces waste test-time compute, and the solution, Guided-ReST, that injects optimal-solution “landmarks” during self-training. The experiments are clear and show measurable gains in Countdown and mixed results with math benchmarks.
- The paper improves test-time compute efficiency by reducing the unnecessary wasteful exploration of mistakes. The method is a training algorithm that encourages more economic behaviour at inference time.
- Although the results are not as compelling on the math benchmarks, the initial effort and implementation of the idea in such benchmarks is commendable.
- Generally well-written and clearly articulated. The paper situates itself nicely within prior work, offering a concise recap of SoS, JSRL, ReST, etc. that clarifies the baseline mechanics and helps the reader contextualize the improvements. Additionally, the paper follows a nice cycle of introducing a claim and then validating it with simple and clear-to-understand results throughout the paper.
- Table 1 is particularly insightful and provides a clear empirical motivation: it reveals that while partial-solution hints boost success rates, they simultaneously lower likelihood—highlighting a distribution-shift problem that explains why naïve distillation fails and thereby setting the stage for the more principled Guided-ReST approach.
- operation-level MDP for improved credit assignment due to long search traces.
- Figure 3 shows the model maintains performance on unseen target numbers and continues to improve as the token budget increases, indicating it has learned transferable search strategies rather than merely memorizing the injected landmarks.
- Ablation in Figure 4 demonstrates that Guided-ReST pre-alignment is essential for PPO to realize its performance gains. Similarly, Figure 7 shows naive subgoal rewards help only marginally, strengthening the case that data-centric Guided-ReST is the main lever.
- Upfront acknowledgement of limitation, i.e., dependence on optimal solutions, is noteworthy.



Weaknesses:

- As acknowledged by the authors, the method—and much of the empirical lift—relies on having exact optimal traces for every training example. The authors themselves concede this as a limitation and propose a stronger “teacher” as a future remedy, but without such oracles, the pipeline is hard to replicate in domains where gold CoT traces are scarce or noisy.
- While accuracy continues to rise up to 4k tokens, there is no accounting of *actual* latency or compute cost. Compute-efficiency claims are based on token budgets, not wall-clock or FLOPs. PPO fine-tuning and operation-level value networks introduce extra parameters and training epochs, so the headline “efficient search” narrative remains unquantified in resource terms.
In short, the pretraining part seems comparable to SoS, but there are extra sub-goal-augmented traces (for 3 iterations) + an operation-level value network, which adds more parameters and FLOPs. Therefore, an account of training FLOPs/GPU-hours would be helpful to substantiate the overall compute efficiency (as opposed to only considering the test-time efficiency).
- Related to the above point, only top-1 accuracy is reported; no look at trajectory length, failure modes, or calibration of the value network. Without this, we can’t tell if the model simply finds more solutions or just spams longer traces that eventually hit a solution. Median / percentile trace lengths could shed light on whether the model spams many tokens at larger budgets compared to the baseline or not.
- Not a reason to reject, however, benchmarking with stronger baselines, e.g., 7-13B parameter open models, would be helpful to show that the gains are not due to originally under-performing models. I understand the GPT-2 experiments follow SoS, and the gains are measurable in that regime; however, the marginal improvement in math benchmarks with larger models calls for scrutiny over the actual gain of the method on some other task in which gains can be demonstrated when large models are used.
- Generality outside arithmetic search is only modestly demonstrated. On MATH-500 and AMC23, the lift over ReST is small and absolute accuracy remains low (≤60 % and ≤32 %, respectively). Moreover, the adaptation drops the elegant subgoal-replacement machinery in favor of a simpler “mask incorrect solution” heuristic, suggesting the method’s sophistication may not carry over cleanly to richer natural-language reasoning. Either a narrower scope claim (effective for tasks with verifiable sub-steps) or a stronger transfer experiment is needed for the main claims to be acceptable.
- It would be helpful if the link between trace quality and accuracy could be probed. The authors argue that higher-likelihood traces improve learning, but beyond Table 1, this “trace noise” is not quantified or correlated with performance (this also is not a major reason for rejection).
- Only average results reported—no error bands (despite marking yes in checklist question 7). A more accurate representation of the results and their variability is required to substantiate the claims, especially in Countdown, as the gains in math benchmarks are quite marginal.

---

> ### Author Rebuttal · Authors · 2025-07-30
>
> Thank you for your helpful feedback. We appreciate your positive remarks on the clarity of our manuscript and your suggestions regarding compute efficiency, pass@k accuracy, and other evaluation metric, all of which help us to strengthen our paper. Our responses to your specific questions are provided below.
>
> ---
>
> **Q1. Reliance on exact optimal traces**
>
> We acknowledge that the reliance on exact optimal traces makes our method not readily applicable to general reasoning tasks. Nevertheless, we believe the paper offers useful insight into two fundamental questions.
> 1. Can outcome-level rewards alone improve an LLM's in-context search capabilities?
> 2. If outcome rewards are insufficient, what additional supervision is needed?
>
> Our results indicate that neither ReST nor PPO on their own discover new search behavior; the model acquires novel search capabilities only when step-by-step guidance is provided through supervised learning. This is evident in the pass@k metrics shown below (please refer to Q3), where ReST improves pass@1 but lags behind the base model from pass@8 onward, whereas Guided-ReST continues to improve across all k.
>
> We plan to explore how this strong assumption can be relaxed (e.g. using long CoT models)–and how far that relaxation can be pushed–in future work.
>
> ---
>
> **Q2. Actual latency and GPU-hours**
>
> First, we compared the compute efficiency of ReST and Guided-ReST and report the GPU hours required for data generation and training below. Guided-ReST adds 7 RTX 3090 GPU hours at the data-generation stage, which amounts to less than a 9% increase in total compute–and even less once the much higher throughput of the A100 used for training is considered.
>
> | Method | Data generation | Training |
> | --- | --- | --- |
> | ReST | 20 hours (RTX 3090) | 55 hours (A100) |
> | Guided-ReST | 27 hours (RTX 3090) | 55 hours (A100) |
>
> Regarding PPO, its computational cost is much smaller than a single iteration of ReST or Guided-ReST.
> - For data generation, PPO only uses 25K training samples, while ReST or Guided-ReST uses 200K training samples.
> - For training, PPO performs 11 forward and 8 backward passes for each sample, while ReST and Guided-ReST perform 10 forward and 10 backward passes for each sample as they run for 10 epochs.
>    - 1 forward for computing old_log_prob
>    - 1 forward for computing ref_log_prob
>    - 1 forward for computing value
>    - 2 forward and backward for updating actor and critic * 4 PPO epochs
>
> Therefore, PPO introduces only a marginal overhead to the overall compute budget.
>
> For inference, our method does not invoke the trained critic. Therefore, it introduces no additional computational overhead and its latency scales only with the number of tokens generated.
>
> ---
>
> **Q3. More analysis on accuracy and trajectories**
>
> Could you please confirm whether the "top-1 accuracy" you mention corresponds to pass@1 accuracy? We repeated the experiments on the Countdown task using the more recent Llama-3.2-1B-Instruct model and computed pass@k accuracy on a subset of 1,000 test samples with unseen targets. The results are summarized below.
>
> | Method | Pass@1 | Pass@2 | Pass@4 | Pass@8 | Pass@16 | Pass@32 | Pass@64 |
> | --- | --- | --- | --- | --- | --- | --- | --- |
> | SoS | 0.551 | 0.633 | 0.686 | 0.736 | 0.774 | 0.805 | 0.841 |
> | ReST | 0.601 | 0.652 | 0.700 | 0.732 | 0.763 | 0.788 | 0.809 |
> | Guided-ReST | 0.630 | 0.731 | 0.835 | **0.896** | **0.944** | **0.965** | **0.975** |
> | Guided-ReST+PPO* | **0.734** | **0.793** | **0.839** | 0.883 | 0.903 | 0.925 | 0.941 |
> (* denotes the method presented as Ours in the paper.)
>
> ReST outperforms SoS at pass@1 yet falls behind from pass@8, which is consistent with recent observation that RL mainly boosts the probability of already-likely correct solutions without discovering new ones [1, 2]. In contrast, Guided-ReST improves accuracy at every k. This explains why ReST+PPO yields limited gains whereas Guided-ReST+PPO remains effective. We also conducted an analysis of how many problems that DFS and BFS leave unsolved can be solved by our methods (Figure 4 in the supplementary material), which further highlights and clarifies the strengths of our approach.
>
> Regarding trajectory length, the table below reports the quantile statistics of the search traces for 10,000 test samples with unseen targets.
>
> | Method | 0.0 | 0.1 | 0.2 | 0.3 | 0.4 | 0.5 | 0.6 | 0.7 | 0.8 | 0.9 | 1.0 |
> | --- | --- | --- | --- | --- | --- | --- | --- | --- | --- | --- | --- |
> | SoS | 197 | 200 | 467 | 664 | 1503 | 2123 | 3874 | 4096 | 4096 | 4096 | 4096 |
> | ReST | 197 | 259 | 518 | 777 | 1570 | 2400 | 3465 | 4096 | 4096 | 4096 | 4096 |
> | Guided-ReST | 197 | **197** | 394 | 653 | 1392 | 1928 | 2998 | 4096 | 4096 | 4096 | 4096 |
> | Guided-ReST+PPO* | 197 | **197** | **332** | **529** | **1127** | **1813** | **2619** | **3582** | 4096 | 4096 | 4096 |
> (* denotes the method presented as Ours in the paper.)
>
> Guided-ReST and Guided-ReST+PPO consistently produce shorter traces than the baselines, whereas ReST sometimes generates longer traces than SoS. This confirms Guided-ReST does not simply generate overly longer traces that eventually stumble into a solution; instead, it conducts a more efficient and effective search.
>
> Thank you for the suggestion and we will incorporate these analyses into the main paper.
>
> ---
>
> **Q4. Link between trace quality and accuracy**
>
> Thank you for pointing this out. We conducted experiments to control the quality (likelihood) of the subgoal-augmented search traces. In Algorithm 1, line 4 lets us control how much of the search trace is pruned. Concretely, we can modify line 4 in two ways:
> 1. Aggressive pruning: replace the current node with the first child visited (e.g., node 2 in Figure 2). This removes the entire exploration history of that subtree, so the resulting traces are expected to be shorter but lower in quality.
> 2. Conservative pruning: replace the current node with the last child node visited (e.g., node 9 in Figure 2). This preserves most of the previously explored trace, so the resulting traces are expected to be longer (inefficient) but higher in quality.
>
> We report the average likelihood and token length of subgoal-augmented search traces generated by each strategy below; as expected, the more aggressive the pruning, the shorter and higher-likelihood the resulting traces. Note that optimal paths are typically around 200 tokens long.
>
> | Strategy | Likelihood | Token length |
> | --- | --- | --- |
> | Aggressive | 0.170 | 759.4 |
> | Original* | 0.102 | 1449.2 |
> | Conservative | 0.078 | 2038.5 |
> (* denotes the method presented as Ours in the paper.)
>
> We also report the performance of Guided-ReST under each strategy with 95% confidence intervals.
>
> | Strategy | Accuracy (seen) | Accuracy (unseen) |
> | --- | --- | --- |
> | Aggressive | 0.6677 &plusmn; 0.0014  | 0.6551 &plusmn; 0.0025 |
> | Original* | **0.6900 &plusmn; 0.0076** | **0.6721 &plusmn; 0.0029** |
> | Conservative | 0.6708 &plusmn; 0.0041 | 0.6464 &plusmn; 0.0101 |
> (* denotes the method presented as Ours in the paper.)
>
> These results highlight that high-quality, efficient search traces are crucial for optimal performance.
>
> ---
>
> **Q5. Confidence intervals**
>
> We ran each experiment on the Countdown task with three independent seeds and report the corresponding 95% confidence intervals below, which demonstrate very low variance across seeds.
>
> | Method | Accuracy (seen) | Accuracy (unseen) |
> | --- | --- | --- |
> | SoS | 0.5747 | 0.5342 |
> | ReST | 0.6282 &plusmn; 0.0019 |  0.5989 &plusmn; 0.0084 |
> | Guided-ReST | 0.6900 &plusmn; 0.0076 | 0.6721 &plusmn; 0.0029 |
> | Guided-ReST+PPO* | **0.7477 &plusmn; 0.0053** | **0.7330 &plusmn; 0.0073** |
> (* denotes the method presented as Ours in the paper.)
>
> ---
>
> **Q6. Large model on different task**
>
> As noted in Q3, we re-ran the Countdown experiments using the more recent, larger Llama-3.2-1B-Instruct model and found that its results were consistent with GPT-2, as shown in the table below.
>
> | Method | Accuracy (seen) | Accuracy (unseen) |
> | --- | --- | --- |
> | SoS (GPT-2) | 0.5747 | 0.5342 |
> | SoS (Llama) | 0.5718 | 0.5391 |
> | ReST (GPT-2) | 0.6279 | 0.6003 |
> | ReST (Llama) | 0.6330 | 0.6033 |
> | Guided-ReST (GPT-2) | 0.6823 | 0.6698 |
> | Guided-ReST (Llama) | 0.6791 | 0.6659 |
> | Guided-ReST+PPO* (GPT-2) | 0.7507 | 0.7361 |
> | Guided-ReST+PPO* (Llama) | 0.7561 | 0.7420 |
> (* denotes the method presented as Ours in the paper.)
>
> Regarding a new task, We are exploring an additional experiment that applies our method to a code-editing setting. While we may not have enough time to complete this study within the rebuttal period, we will run it regardless to strengthen our paper. If the results are ready before the rebuttal period ends, we will share them with you.
>
> ---
>
> Once again, we really appreciate your insightful questions, which greatly help us to improve our paper. We hope that our response above addresses all of your questions.
>
> ---
>
> **References**
>
> [1] Yang Yue et al., Does Reinforcement Learning Really Incentivize Reasoning, arXiv 2025 \
> [2] Andre He et al., Rewarding the Unlikely: Lifting GRPO Beyond Distribution Sharpening, arXiv 2025

---

> > ### Comment · Reviewer_xv2X · 2025-08-08
> > **Response to Author Rebuttal**
> >
> > Thank you for the additional results and explanations, which addressed most of my concerns. I have therefore increased my rating.

---

> > > ### Author Response · Authors · 2025-08-09
> > >
> > > Thank you for acknowledging our rebuttal. We are glad to hear that most of your concerns have been addressed. We greatly appreciate the time you spent providing a detailed review with various suggestions for analysis, and we will ensure that the revised manuscript includes the new experimental results accordingly.

---

### Note · Authors · 2025-08-14

Dear Area Chair and Reviewers,

We are sincerely grateful to the Area Chair and Reviewers for their invaluable time and constructive feedback. In response to the reviews, we made the following key improvements, which we believe have substantially strengthened our paper:

1. Generalization: We addressed concerns about our GPT-2 model by conducting new experiments on a modern Llama architecture. The consistent performance gains confirm that our method is not limited to smaller models.
2. In-depth analysis: We provided a much richer evaluation, including pass@k accuracy, trajectory length statistics, and 95% confidence intervals, to further quantitatively support our claim of improved search efficiency and effectiveness.
3. Enhanced clarity: We improved the overall clarity of the paper by refining our problem formulation and methodology, providing a clearer explanation of our core contribution, and adding a detailed discussion of our method's limitations.

We are particularly encouraged that all four reviewers found our rebuttal convincing enough to raise their scores. We acknowledge the remaining concern regarding the need for more diverse benchmarks, which is a valuable direction for further strengthening our work. We will incorporate all promised revisions from our discussions into the camera-ready version.

---

### Decision · Program_Chairs · 2025-09-17

**Decision:**

Accept (poster)

**Comment:**

This paper proposed a new fine-tuning algorithm to improve the search capability in inference time of LLMs. The topic is well motivated and the paper well written and easy to follow. There are some limitations of the proposed method: it depends on having access to optimal solutions, and the main benefits are primarily demonstrated on the Countdown tasks. Despite this, most of the reviewers found this paper and the preliminary results interesting. We recommend acceptance of this paper, but encourage the authors to add more empirical investigations w/o direct access to optimal solutions and other math / reasoning tasks in the final revision.